# Development and Validation of Safe Motherhood-Accessible Resilience Training (SM-ART) Intervention to Improve Perinatal Mental Health

**DOI:** 10.3390/ijerph20085517

**Published:** 2023-04-14

**Authors:** Shireen Shehzad Bhamani, David Arthur, An-Sofie Van Parys, Nicole Letourneau, Gail Wagnild, Olivier Degomme

**Affiliations:** 1School of Nursing and Midwifery, Aga Khan University, Karachi 74800, Pakistan; 2Department of Public Health and Primary Care, Faculty of Medicine and Health Sciences, Ghent University, 9000 Ghent, Belgium; ansofie.vanparys@ugent.be (A.-S.V.P.); olivier.degomme@ugent.be (O.D.); 3School of Nursing, Peking Union Medical College, Beijing 100005, China; davidgordonarthur@gmail.com; 4Faculty of Nursing, University of Calgary, Calgary, AB T2N 1N4, Canada; nicole.letourneau@ucalgary.ca; 5Resilience Center, Montana, MT 59101, USA; gwagnild@resiliencecenter.com

**Keywords:** resilience, pregnancy, perinatal maternal health, mental health intervention, Pakistan

## Abstract

Perinatal mental health issues in women can lead to a variety of health complications for both mother and child. Building resilience can strengthen coping mechanisms for pregnant women to improve their mental health and protect themselves and their children. The study aims to develop and validate the contextual and cultural appropriateness of the Safe Motherhood-Accessible Resilience Training (SM-ART) intervention for pregnant women in Pakistan. A three-phase approach was used to develop and validate an intervention that promotes resilience in pregnant women. Phase I comprised a needs assessment with stakeholders (pregnant women and key informants) to elicit opinions regarding module content. In Phase II, an intervention to build resilience was developed with the help of a literature review and formative assessment findings, and Phase III involved the validation of the intervention by eight mental health experts. The experts assessed the Content Validity Index (CVI) of the SM-ART intervention on a self-developed checklist. The resultant SM-ART intervention consists of six modules with strong to perfect CVI scores for each of the modules. Qualitative responses endorsed the strengths of the intervention as having innovative and engaging activities, contextual and cultural relevance, and a detailed, comprehensive facilitator guide. SM-ART was successfully developed and validated and is now ready for testing to promote the resilience of pregnant women at risk of perinatal mental illness.

## 1. Introduction

Perinatal mental illnesses such as anxiety, depression, bipolar disorder, and postpartum psychoses have a significant impact on maternal and infant morbidity [1,2,3]. Meta-analyses of studies conducted on samples of pregnant women reveal that one-fifth of women experience a mental health issue during their pregnancy [4,5,6] with 10–20% of women encountering symptoms of depression or anxiety, or both depression and anxiety [7]. A systematic review and meta-analysis of forty-three studies revealed that in Pakistan, the pooled prevalence of antenatal depression and postnatal depression were 37% and 30%, respectively [8].

Perinatal mental illnesses are strongly associated with a range of physical (congenital malformations, undernourishment, low birth weight, and stunting) problems in infants [9,10,11,12,13]. For example, a retrospective Australian study concluded that infants of depressed pregnant women had lower birth weight and shorter gestational ages, whereas significant birth complications were noticed in babies of women who experienced anxiety during pregnancy. Low Apgar scale scores (a 1–10 scale representing how well the baby is doing after birth, with 10 being the best) were common in both the depression and anxiety groups [7]. Evidence of decreased cephalic circumference and challenging cognitive issues in infancy is also related to mothers’ disturbed mental health during pregnancy [14]. Children born to mentally disturbed mothers can face numerous developmental issues related to cognition, speech, and behavior [15], as well as autism and attention deficit disorder [16]. Epigenetic research explains the association between the poor mental health of mothers and their children. Babies born to such mothers have emotional dysregulation due to disturbed fetal programming and increased stress sensitivity [17]. A study by Send et al., 2014 [18] showed that maternal stress and anxiety are negatively correlated with fetal longevity. Moreover, telomere length (a proxy for biological aging) is significantly shorter in a fetus with a depressed mother compared to a relatively healthier mother [18].

Perinatal mental illness and coinciding elevated levels of cortisol in the perinatal period can lead to a variety of mental and physical health problems in mothers and their children. Issues such as pre-eclampsia, premature and difficult births, increased risk of fetal growth restriction, and bonding and attachment problems can all be directly correlated with mental health problems during pregnancy [19]. As these issues can lead to lifelong health concerns that span across generations, the World Health Organization strongly emphasizes the need to allocate greater attention to the prevention of mental illness during pregnancy and the promotion of mental health in healthcare systems across the globe [20]. Many organizations are helping create a space where expectant mothers can express their worries [21,22,23].

During the perinatal period, women who demonstrate resilience attributes such as moderate to high levels of self-efficacy, sense of mastery, self-esteem, and optimism can deal effectively with stressors in the environment [24]. The concept of resilience varies among authors. One simply states that it is an ability to cope with shocks and continue life as a routine [25]. Southwick et al., 2012 [26], on the other hand, described it as an ongoing process of positive adaptability regardless of the hardships of life [26], wherein it is viewed as a buffer to stress and a sign of greater emotional regulation [27,28]. Resilience has also been considered a personality trait, although it is currently viewed as a learned skill that develops with the adversity of life [29]. Higher resilience is associated with reduced anxiety and depression [30], and women who possess resilience traits are better able to handle environmental challenges. Low resilience has been associated with poorer pregnancy outcomes, and higher resilience increases coping ability [31]. One study highlighted resilience as a mediating factor in smooth pregnancy [28], while another study conducted in Spain examined hair cortisol levels and found an inverse relationship between resilience and cortisol levels (an indicator of stress) [32].

Thus, promoting perinatal mental wellbeing via enhancing pregnant women’s inner strength, building resilience, and adopting healthy coping skills could prevent and address perinatal mental illness [33,34]. Such strategies show promise as early interventions, for mothers and their children’s mental well-being, are being based on a paradigm shift from a disease-based to a strengths-based approach.

Previous studies have shown that Cognitive Behavioral Therapies (CBT) [35] and Positive Psychology interventions [36] are effective in enhancing the mental wellbeing of women during the perinatal period. A trial conducted in Iran used CBT as an intervention to reinforce mental wellbeing during pregnancy. Scores for depression, anxiety, and stress were found to be significantly lower in the intervention group compared to the control group [37]. Similarly, another study conducted in Turkey reported that CBT during pregnancy results in a decline of post-intervention anxiety scores [38]. A RCT conducted in Australia on mindfulness intervention (meditation and cognitive exercises) to reduce depression and anxiety in the perinatal period reported significant decreases in depression, anxiety, and stress scores in the intervention arm [39]. To date, there was only one trial conducted in Pakistan that tested the effectiveness of a ’thinking healthy’ program in pregnancy; however, no change was observed post-intervention [40]. Yet to our knowledge, interventions enhancing resilience to prevent and/or address perinatal mental illness have neither been created nor tested for pregnant women in Pakistan. Hence, resilience, or the capacity for strength, positivity, optimism, competency, and capability to be able to cope with daily life stressors, may be a useful focus for pregnant women [41].

The pregnancy period is considered a window of opportunity to foster resilience and adaptation in mothers and their newborns [42]. Programs that promote resilience in the perinatal period aim to improve mental well-being in a way that is individual-focused, cost/time effective, and encourages self-motivated practices [35,36]. Our literature search shows that there is a scarcity of research on positive psychological interventions in Pakistan, especially for pregnant women. We expect to bridge this gap by adding to the evidence-based literature on enhancing the individual strengths and positive mental well-being of women during their perinatal period, through the development and validation of a strengths-based resilience-building module. This paper reports the development and validation of the Safe Motherhood-Accessible Resilience Training (SM-ART) intervention for pregnant women in Pakistan.

## 2. Materials and Methods

### 2.1. Study Design

The study was conducted in three phases: Needs Assessment; Development of Resilience Building Intervention; and Validation of Intervention. Ethics approval (ERC-2020-1197-10212) was obtained from the Research Ethics Committee of the Aga Khan University, Pakistan, before the commencement of data collection. Written consent was obtained from all participants.

#### 2.1.1. Phase I: Needs Assessment

To develop the intervention’s content, a review of the literature and interviews with 25 pertinent stakeholders (17 pregnant women and 8 key informants, including psychologists, psychiatrists, and nurses) were conducted.

All the women were married and aged 18 years or older, with a 12-week and above gestational age. The key informants were mental health experts with more than five years of field experience. All the experts were well-versed in the Pakistani cultural and clinical context. Moreover, both government and private representation were ensured while recruiting.

The semi-structured interview questions ascertained pregnant women’s views on the role of resilience-building skills and their attribution to the management of mental health issues. They were asked about their own experiences managing everyday challenges. The interview guide developed especially for these women has explored the resilience attributes incorporated within Wagnild and Young’s resilience framework [43,44]. The concepts explored during the interview were purpose/meaning of life; perseverance/determination; existential aloneness/friendship with oneself; equanimity; and self-reliance. On the other hand, KIs interview questions focused on their opinions about the module’s content, structure, and approach as well as their opinions about its applicability/relevance in a Pakistani context. They were asked for their availability to validate the intervention after its development. Both the stakeholders (participants and KIs) were asked about the acceptability of the proposed intervention. Their interview guide was developed by the research team after a thorough literature review and based on their prior experiences.

In order to avoid outside noise and distractions, interviews with key informants were conducted and recorded in a private space. The facilitator kept field notes to record the participants’ non-verbal responses—such as eye contact, facial expressions, tone and pitch of voice, and body language. The interviews with the pregnant women lasted between 40 and 60 min, and 35 and 50 min for the mental health experts. Interview sessions were continued until data saturation was reached—that is, when the participants delivered little or no new information.

The responses that were conducted and transcribed into Urdu were translated into English and then translated back into Urdu. The back-translation into Urdu was compared with the original transcription to establish coherence between the English and Urdu versions of the interview and to detect any discrepancies in translation. The data were manually analyzed using thematic analysis. Verbatim transcriptions were coded openly and merged into categories and themes. Themes were generated, and each theme was named using a content-characteristic description. For data analysis, a variety of sources (field notes, audio recordings, transcriptions, and the researcher’s reflections) were considered. The detailed analysis steps we used are described in our initial project publication [45].

#### 2.1.2. Phase II: Development of the Intervention

It emerged from Phase I that an intervention would need to be adaptive and engaging given the variety of issues faced by Pakistani women, including cultural factors, domestic responsibilities, academic difficulties, dropout risk, husband approval, and a lack of enthusiasm for learning. Additionally, Phase I findings highlighted the need to create a comprehensive yet accessible approach because of the anticipated variety of personal characteristics related to educational background, ethnicity, age, and gestational age. Therefore, the research team members having expertise in mental health, maternal and perinatal health, and intervention development have carefully and thoroughly reviewed, discussed, and developed the content of the intervention based on the feedback and responses from stakeholders. The consensus from numerous team discussions led to the development of contextually appropriate activities and approaches. The principal investigators explored teaching and learning engagement strategies appropriate for the target audience. The content of each module was delivered through a combination of multiple innovative and creative strategies (reflective stories, culturally relevant pictures, eye-catching flashcards, soothing poetry, interactive discussion, roleplay, etc.) on the theme. This is to avoid a monotonous approach and ensure that all participants can understand and feel involved, engaged, and retained. We, therefore, want to create material and have also worked on a method for delivering and communicating the content in the most appealing and effective way.

Considering the low literacy rate of Pakistani women, the Gunning Fog Index (GFI) was applied to each sentence, paragraph, and module to verify that the language used was simple, easy, and comprehensive [46]. The GFI employs a six-step process to extract a quantitative measurement of the grade level of education to which a specific section of text pertains. Specifically, sentence and word length are calculated, and it is recommended that the index score be in the single digits in order to increase readability and accessibility [46]. This index allows us to scientifically prove that our intervention is understandable.

In order to use a more stringent process to test quality, which was only available in English, the module was developed in English before being translated into the local Urdu language by a professional linguist.

The intervention comprised six modules based on six different themes. Each theme contained innovative activities designed sequentially. Comprehensive directions and systematic guidelines were developed for facilitators to ensure the consistent conduct of the modules. It also contains all the training resources that can be used by trainers. Each module was structured in the same way, including the aim, learning objectives, required material according to the list of activities, and notes to aid facilitators in communicating the content and minimize facilitator variances.

#### 2.1.3. Phase III: Validation of the Intervention

To validate the appropriateness and relevance of the intervention, a panel of professional mental health experts was assembled. Our expert panel included two mental health nurses, two psychiatrists, two psychologists, and two maternal and child health experts. The selection of these eight content experts was performed deliberately to capture diverse feedback based on their different specializations [47].

Based on a thorough review of the literature [48,49,50,51,52,53], a 15-item checklist was developed, which included four items on Content and Resource Material, two items for Illustrations, and three items each on Scientific Accuracy, Legibility and Printing Characteristics, and Literary Presentation. The checklist was given to the eight experts to rate each module on a scale of 1 for “strongly disagree”; 2 for “disagree”; 3 for “agree”; and 4 for “strongly agree”. The content validity index (CVI), which is the most widely utilized method for content validation, was calculated using the following formula: I-CVI = (agreed item)/(number of experts), and S-CVI/Ave = (sum of I-CVI scores)/(number of items). The CVI ranges from 0 to 1, representing the measure of agreement between experts. A CVI value > 0.83 is considered acceptable [54].

To learn more about the areas that require further improvement, two open-ended questions were added to the checklist. The experts, who agreed to take part, were sent a Google Drive link with all the necessary materials via email. These resources included their consent and a brief description of the project, a Google Form link with a checklist for the content’s relevance and clarity, a SM-ART intervention, and a Voiceover PowerPoint presentation (VOPP) to help participants understand the intervention more thoroughly. The participants were invited to share their feedback within 4 weeks.

## 3. Results

### 3.1. Phase I: Need Assessment

Seventeen pregnant women from diverse socio-economic backgrounds were enrolled. Six themes emerged from their in-depth interviews as a part of the SM-ART intervention. These include: Finding the Purpose in life; Dealing with Emotions; Believing in Yourself; Adapting an Optimistic Approach; Strengthening Support Systems and Relationships; and Internalizing Spirituality and Humanity. Details about themes, the content of the intervention, and the socio-demographic characteristics of these participants are published in another paper on this project [45].

A total of eight KIs (four mental health nurses, two psychologists, and two psychiatrists) were interviewed. The mean age of KIs was 36.37 years (SD ± 6.72). Fifty percent of the KIs were literate up to the master’s level, and the remaining had either a bachelor’s (25%) or doctorate (25%) level education. They had an average of ten years of work experience, with a range of 6–18 years. Notably, all KIs were female, with 62.5% married, 25% single, and 12.5% divorced. The majority of the KIs were employed in the private sector (62.5%). Findings from the KIs interviews were congruent with those of the pregnant women in terms of intervention content like coping strategies, optimism, interpersonal communication skills, social support, emotional regulation, self-awareness, self-worth, relationship management, sense of humor, and hope. They also included spirituality, helping and supporting others, and self-reflection.

Regarding the acceptance of the intervention, all participants recommended that it be carried out in a casual, participative, and relaxed setting to promote the greatest possible interaction and involvement between participants and the facilitator. Some also express their lack of desire to pursue further education, having been removed from the university environment. Most of them expressed a preference for group training. Several potential barriers to the intervention were highlighted relating to Pakistani culture, including getting family approval (from husbands and in-laws); managing household duties and childrearing to make time for sessions; transportation costs for commuting; and finding childcare. However, they also provided some alternatives, suggesting that if childcare, food, and transportation were provided, more people might be persuaded to attend sessions.

### 3.2. Phase II and III: Development and Validation

The in-depth interviews with participants resulted in six themes that became the basis of our intervention development. Results from the pregnant women’s interviews were published earlier [45].

The SM-ART intervention takes the form of a booklet with front and back covers, a table of contents, descriptions of activities for each module, and the facilitator’s resources. A significant effort went into the design of the cover page, first page, and last page of the intervention (see Figure 1).

All images were drawn by hand by a hired illustrator with public health nursing experience. On the first page, the pregnant woman holding an umbrella symbolizes how SM-ART training can assist women in coping with all types of hardships and extreme weather, and on the last page, it shows how, after six weeks of training, the entire house or family can be empowered with resilience through one woman’s education. The facilitator’s guide provides direction for the conduct of the exercises as well as potential responses to discussion questions after each activity. The intervention was designed to be delivered in a group setting to give participants a relatively secure and safe space in which they could learn about and discuss the concepts, express their thoughts and experiences, and give and receive support from one another. Each module was delivered once a week with a duration of 60 to 90 min and an overall of 8.5 h in total. This was based on the needs assessment, which showed participants preferred weekly sessions that lasted no longer than 1.5 to 2 h. Each module is unique in its purpose, distinct from the others, and can stand alone. Two activities are shared across all modules: (1) the meet and greet period with a mood assessment that was expressed using emoji stickers (excited, happy, confused, sad, and anxious) before and after each module; and (2) module evaluation at the conclusion of the session.

Table 1 provides an overview of the module theme, purpose, objectives, activities, and time duration of each module of the SM-ART intervention, while Table 2 details each of the activities targeted to achieve these module objectives.

Two or three rounds of GFI were conducted for each module. In the first round, modules with a GFI greater than 13 were targeted. The index of individual sentences was calculated, and specific words were changed in the sentences. These changes were then applied to the full transcript of the page. In the second round, changes within the transcript were made rather than targeting individual sentences. Recurring words were changed to synonyms that were simpler to understand or removed if unnecessary. Trial two decreased the index to around seven or eight; if this was not achieved, a third round was conducted. This round included revisiting the transcript and determining if sentences or phrases could be shortened without losing their meaning. The index calculations after the final trial varied between modules, with the highest index being nine and the lowest index being seven (Table 3).

Out of the 12 content experts who were approached, eight of them participated in the validation of the SM-ART intervention. The mean age of participants was 42.5 years (SD ± 4.98), with a range of 36 to 53 years. Most of the experts held a doctorate (62.5%), and the rest had a master’s degree (37.5%). There were six experts from within the country and one each from Australia and Canada. All were originally from Pakistan and have an average of 14.5 years (SD ± 7.2) of working experience in Pakistan (range: 5–25). The total number of years of their field experience ranges from 10 to 25, with an average of 17.5 years (SD ± 5.4). The CVI scores for modules and their components ranged from very high to perfect (Table 4).

Additionally, thoughts and suggestions regarding the intervention, in general, were requested from the experts. Amendments were made to some aspects of the evaluation, such as general formatting and paragraphs, sentence structure, and the proposed duration of the training module. A limited number of categories were discovered from the two open-ended questions (strengths and suggestions). The written statements of the experts support the findings of intervention strengths.

### 3.3. Content Expert Feedback

#### 3.3.1. Innovative and Engaging Activities

One of the strengths highlighted by the experts was the module activities. The content experts felt that the activities that were included were well conceived, engaging, creative, innovative, and well-connected with the purpose. In the experts’ words: “All are activity based and seem engaging. It will provide an opportunity to the participants to share their thoughts and ideas.”…“I must appreciate the creativity”…“I am very impressed how activities are articulated”…“…the activities, simple yet serve the purpose”…“All the activities are aligned to the topic”…“Activities are creatively designed”…“This module is very impressive, especially the Graffiti wall of coping”.

#### 3.3.2. Contextually and Culturally Relevant

Overall, the experts believed that everything was culturally and contextually relevant to the population for which the intervention was designed. Responses included: “The relatability to our local context and different activities which keeps the whole experience engaging”…”local examples of daily living are best to make it understanding and relatable”…“Wisely designed module with the cultural norms and values”…“Each image/ picture has been carefully drawn while taking in mind Pakistan’s setting and culture”.

#### 3.3.3. Comprehensive Facilitator Guide

The facilitation guide was seen as a valuable asset by some experts. According to them, the facilitators and trainers should greatly benefit from the instructions and guidelines offered in each module, as they will help to standardize the intervention. Comments included. “To prevent errors and ensure consistency, the module includes very specific instructions for facilitators” and “It’s great to read the thorough guide, and the facilitators’ notes are very useful in these modules”.

#### 3.3.4. Consideration of Different Learning Styles

The experts appreciated that the authors considered various learning styles when preparing activities. One expert “loved the combination of activities in each model. I am sure, this will be useful for all types of learners”, another appreciated “the careful planning that went into the preparation of the activities. Every type of learning (auditory, visual, etc.) was considered”.

#### 3.3.5. Formatting and Language

Some of the content experts’ comments focused on the modifications of language and overall formatting for better understanding, organization, and clarity. For example: “Possibly, language could be made even simpler” …“objectives could be simplified further for better understanding”…“Font of a poem in Urdu can be changed, for headings, consistent font size and style should be used”.

#### 3.3.6. Content and Delivery Recommendations

The experts provided some recommendations for the module content and its execution in the future. Their comments included: “In module 5, connect lizard story with ‘rescue’ fantasy concept”…“In the truck of relationship, it can also be interpreted as taking turns (e.g. driver or passenger) ensure sustaining and strengthening relationships”…“In all the modules, you will need to keep time for feedback”…“In ‘Gup shup’ area, you can place a whiteboard or chart where participants can write (if they can) regarding their immediate reaction or feeling”…“I would suggest that you should plan to test the effectiveness of each module”.

## 4. Discussion

The major outcome of this study was the development and validation of the resilience-building intervention SM-ART, which was designed for pregnant Pakistani women. To ensure face, content, and cultural validity, qualitative data were obtained from stakeholders and pregnant women, while ensuring that this tailored intervention was consistent with socioeconomic status and the context of the participants’ lives. To be ‘state-of-the-art’, a formative phase is considered essential—paying attention to stakeholders concerns so that they can be addressed in the development of the intervention [55]. For example, there are high levels of illiteracy in Pakistan, so many people are unable to comprehend written health-related material [56]. Understanding the underlying issues and being mindful of the limited availability of mental health care for pregnant women, the SM-ART intervention encompasses rigorously designed modules by creating an Urdu version with simplistic language to increase the degree of comprehension and address the range of likely occurring mental health issues with appropriate group and self-care interventions.

Using the GFI increases the rigor of the language content by challenging initial perspectives about which phrases and words are generally used and ultimately facilitating communication skills useful in participants’ daily lives. Gills 2012 [57] advocate this index as a solution to the accessibility problems of language faced by the public, as it builds the capacity of a person to find access to, contextualize, and comprehend the information required for making healthcare decisions [57]. Insights from the Gunning Fog process included the accessibility of storytelling, the removal of unnecessary words, and application to daily life. The stories within the modules always had lower and better GFI scores compared to the instructions or insights because stories often had simple dialogue and were straight to the point, indicating that they were an accessible form of transmitting the information.

Additionally, CVI is an important measure for validating the results and assessing the appropriateness and practicability of the intervention [3,58]. The validation of our SM-ART modules has good to perfect CVI scores, which is evident from the high applicability and relevance of this intervention.

In line with the literature, which advocates involving experts at the development and validation stages, this study involved experts in the needs assessment phase and the content development, validation, and accuracy. The participation and involvement of experts at every level contributed to high to perfect CVI scores and increased the relevance of the content.

The methodology and adaptation based on the suggestions of the reviewers allowed the module to be improved and tailored to the needs of the target population. Additionally, SM-ART contains appealing language, eye-catching images, and engaging activities that may match the reader’s mindset and improve readiness to learn. Research shows that using a variety of strategies for engaging participation, helps participants stay focused, understand concepts, have productive debates, and behave properly as a group [59]. Another study that looked at adult learners found that there should be more actions than just listening to engage participants and make them active learners. They must engage in discussion, debate, writing, problem-solving, and higher-order thinking activities [60]. Another systematic review emphasized that Resilience Training should also involve experiential learning, which may be accomplished through role-playing, demonstration, different skill-based training, and reflection [61]. Our intervention, therefore, attempted to include a variety of activities to encourage their involvement and commitment.

The essence of SM-ART arises from its cultural relevance to pregnant Pakistani women, its contextual connection with Pakistani society, and its implications for pregnant women. Considering the socio-economic structure of Pakistan, SM-ART traverses the everyday problems faced by pregnant women and how these adversely affect their mental and physical health. Learning from this, SM-ART creates a set of intuitive and appealing activities that address the issues these women might face. Research shows that interventions that are understandable and culturally appropriate, have the best likelihood of being effective [62].

In addition, the suggested face-to-face approach of the sessions ensures that there is a high chance of better understanding of knowledge and skills amongst users, and since the frequency of these sessions is only once per week, it also makes sure that these users are not being overburdened in an already demanding stage of their lives. Additionally, doing these sessions in a group may create a sense of community for these women, where they can meet other women facing similar issues and talk out their concerns. This not only creates a support group on which these women can rely but also a place where they can freely express themselves. Studies also found that the most effective preventative measure against prenatal depression is social support [63,64].

Moreover, addressing the language barrier, running numerous field tests with Pakistani women from a variety of social backgrounds, and content validation by expert panels maximize the chances that the SM-ART covers content that is often missed by other sources available either through television, textbooks, or social media. Studies have shown how language barriers can be precursors for inequality in perinatal healthcare, especially among groups of migrant women for whom the native language is a foreign one [65]. Similarly, a large part of the Pakistani population, specifically those in Karachi, are migrants with an array of different cultures and languages. This high diversity meant that it was essential to use one commonly understood language (i.e., Urdu). Another study conducted in Pakistan concluded that to maximize the effect of healthcare teachings, it was imperative to keep into account the social settings and the cultural relevance [66]. Therefore, SM-ART was developed with a strong emphasis on maintaining its teachings within the cultural aspects of Pakistan.

The different learning styles adopted by the modules allow users to choose their preferred way of learning: audio, visual, group, or a mix of these styles. Finally, the facilitator’s guide offers a platform for instructors to improve their training delivery while establishing a level of facilitation that is as consistent as possible across the team of instructors. Additionally, this will offer instructions that the general public can utilize to learn to communicate effectively in their daily lives. The experts concurred that the facilitator guide is necessary for these sessions and that it was carefully crafted with the facilitators in mind.

Resilience-building interventions already exist for the following populations: cancer patients [67], school-based adolescents [68], military [69], immigrant children [70], transgender [71], diabetic fatigue patients [72], adolescents with psychiatric symptoms [73], medical residents [74,75], youth at high risk of HIV [61,76], mothers of cancer patients [77], and nursing students [78]. However, there are no resilience-based interventions that are aimed at the perinatal period. The first contextual and culturally appropriate intervention for pregnant women was developed in our study. Moreover, the literature supporting resilience theory strongly supports the themes of our modules, which offers us some assurance that we are moving in the direction of positive psychology [68,77,78,79,80,81,82].

This study is the first of its kind to be conducted in Pakistan. The SM-ART intervention is the result of in-depth research, resulting in a contextually and culturally relevant intervention program. Furthermore, having a panel of experts review the module content ensured the validity of the prepared intervention by allowing for further iterations and refinement. The study also provides a basic framework for any future research in this domain. Moreover, the name Safe Motherhood “ART (Accessible Resilience Training)” may help diminish the stigma associated with mental illness in this Pakistani context. Further, the intervention has a training manual for facilitators to add to enhance reliability over time. In addition, the intervention is also translated into the national language (i.e., Urdu), so it can be used in any area of Pakistan.

Along with its strengths, this study also has some limitations. Firstly, it has limited generalizability as it was only conducted focusing on married pregnant women from a low socioeconomic background. It can be generalized to a population with similar characteristics only, with modifications to the examples and images necessary before it can be applied to a different population, thereby, enhancing contextual relevance for the targeted users. Secondly, the involved KIs and experts were all female, which omitted male perceptions and insights that could be included in future module development.

## 5. Conclusions

This study marks an important step in the development of the novel resilience-building intervention, which consists of six theme-based modules. The SM-ART offers a road map for fostering resilience among pregnant women. A valid resilience-building module can also provide reference material to guide midwives as facilitators of resilience-building during the perinatal period. Future testing in a rigorously controlled research study is necessary to further test the impact of SM-ART as a whole or as a separate module on building resilience, coping, self-worth, and the management of anxiety and depression in local pregnant women. In a healthcare landscape where access to quality mental health care is difficult, if not impossible, this self-care approach is hypothesized to be not only empowering but also effective.

## Figures and Tables

**Figure 1 ijerph-20-05517-f001:**
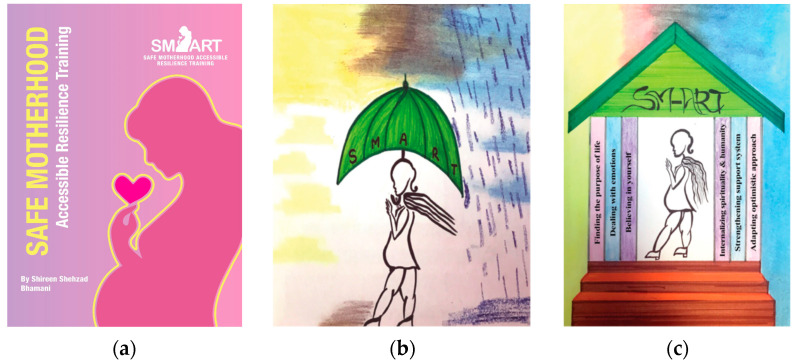
(**a**) The cover page of SMART Intervention; (**b**) The first page of SM-ART intervention; (**c**) The last page of SM-ART intervention.

**Table 1 ijerph-20-05517-t001:** SM-ART intervention modules, objectives, strategies, and time duration.

Module Title Page	Objectives of the Module	Strategies and Activities	Duration
Finding the Purpose of life	1. To understand the significance of having a purposeful life; 2. To reflect on their own goals and purpose in life; 3. To share their life experiences with others and learn from others; 4. To connect the learning from this module to their daily lives; 5. To apply the knowledge received from this module in their lives.	Crack Pot story, Pass the Bowl, Recipe for Biryani Wheel of Life.	60–75 min
Dealing with Emotions	1. To understand the impact of emotions on an individual’s behavior; 2. To reflect on their everyday emotions and behavior; 3. To share their way of dealing with negative emotions; 4. To apply the learned knowledge in their daily lives and to be able to experience positive emotions.	A Rapid Quiz, Complete Me, Jigsaw, Graffiti Expression Wall of coping.	75–90 min
Believing in Yourself	1. To understand their self-worth; 2. To realize their value as an individual and look at the brighter side of life; 3. To reflect on their strengths and abilities; 4. To apply the knowledge and strengths in their daily lives.	Crumpled Rupee, Johari Window, A Coin Has Two Sides, What Describes you Best?	60–75 min
Adapting an Optimistic Approach	1. To understand the importance of an optimistic approach; 2. To reflect on the way they see their lives; 3. To share their life experiences and adopt positivity in their lives; 4. To connect and create a positive environment for themselves and others; 5. To apply and integrate these concepts in their daily lives.	Paste the Positivity, Motivational Stories, Place yourself, Roll the Dice.	60–75 min
Strengthening Support Systems and Relationships	1. To understand the importance of a strong support system; 2. To reflect on their role in dealing with their relationships; 3. To practice those skills that are important in managing any relationship; 4. To apply the knowledge in their daily lives.	My Supporters, Power of Communication, Truck of Relationships.	75–90 min
Internalizing Spirituality and Humanity	1. To understand the role of spirituality and humanity in mental well-being; 2. To discuss the importance of being empathetic and grateful; 3. To reflect on the concepts of humanity and spirituality in their daily lives; 4. To internalize and apply the learned knowledge.	Reflect on the statements, Humanity Comes First, Be Grateful, The Power of Faith and Trust.	75–90 min

**Table 2 ijerph-20-05517-t002:** Modules’ Strategies—purpose and outcomes.

Strategies	Purpose and Outcomes
**Module 1: Finding the Purpose of Life**	
Crack Pot story	Participants learn that everyone, even those who believe they are useless, has a purpose in life. They continue to perform for an unknown purpose.
Pass the Bowl	Participants gain insight into their life’s purpose through this activity. Everything in the universe has a purpose. Given that humans are superior to other species, our purpose in life should be significant.
Recipe for Biryani	Participants consider their own life’s purpose and the things that are significant to them.
Wheel of Life	Participants consider their life goals and what they may do to pursue them. This also considers how essential it is to balance and set priorities for the things that are important to us. like family, health, and taking care of oneself.
**Module 2: Dealing with Emotions**	
A Rapid Quiz	Participants learn to recognize their emotions before trying to manage them.
Complete Me	Participants learn how unexpected events can occur in their daily lives and how theycan impact their emotions. To manage emotions and behave appropriately, a person must be aware of their feelings.
Jigsaw	Participants exchange experiences and learn how other people feel in similar circumstances. It is important to share and learn from one another’s experiences because everyone handles stress differently.
Graffiti Wall of coping	Participants learn different stress-reduction techniques, consider their coping mechanisms, and take up some useful advice.
**Module 3: Believing in Yourself**	
Crumpled Rupee	Participants learn to be aware of their value, which is unchanging regardless of circumstances. They learn to be self-assured and confident in their skills.
Johari Window	Participants learn the self-awareness technique and recognize their relationships with themselves and with others.
A Coin has Two Sides	Participants accept their strengths and weaknesses and learn to consider the positive side.
What Describes you Best?	Participants reflected on their lives ups and downs and how different people handle stress in different ways. They also learn that strength comes from believing in oneself and one’s abilities.
**Module 4: Adapting an Optimistic Approach**	
Paste the Positivity	Participants learn how small things in life can bring hope. All they need is to change the way they look at things.
Motivational Stories	Participants learn to keep a positive attitude and pursue their goals. They also understand how an optimistic approach can overcome all obstacles.
Place yourself	Participants consider their personal approaches to stress. They learn ideas for their daily lives that determine their positivity.
Roll the Dice	Participants understand the positive aspects of their lives with the help of pictures that represent optimism, hope, and enthusiasm.
**Module 5: Strengthening Support Systems and Relationships**	
My Supporters	Participants reflected on their support systems and became aware of their supporters. They understand that neighbors, family members, friends, and other acquaintances play a variety of roles in their lives.
Power of Communication	Participants understand the importance of communication in maintaining and fostering relationships. They also learn strategies to express themselves better.
Truck of Relationships	Participants learn how they can improve their relationships. They also understand that relationships can be difficult at times, but they can improve with effort and optimism.
**Module 6: Internalizing Spirituality and Humanity**	
Reflect on the Statement	Participants understand that there is always a reason behind anything that happens.
Humanity Comes First	Participants comprehend the idea of humanity and learn how making others happy allows them to experience happiness too.
Be Grateful	Participants learn the value of thankfulness and the positive effects that it may have on their lives.
The Power of Faith and Trust	Participants reflect on the concept of faith and learn how it plays a role in their lives and helps them in difficult times.

**Table 3 ijerph-20-05517-t003:** Gunning Fox Index Calculation.

	First Draft	Trial 1	Trial 2	Trial 3
Module 1: Finding the purpose in life	10	9	7	7
Module 2: Dealing with Emotions	13	10	9	8
Module 3: Believing in yourself	10	9	8	8
Module 4: Adapting an Optimistic Approach	11	10	8	8
Module 5: Strengthening Support and Relationships	12	10	9	9
Module 6: Internalizing Spirituality and Humanity	9	8	8	8
Average	11	9	8	8

**Table 4 ijerph-20-05517-t004:** The CVI of each module as per the content experts’ opinions (*n* = 8).

Module	Scientific Accuracy	Literary Presentation	Content and Resources	Illustration	Legibility and Printing	Overall Average
1	0.958	1	1	1	1	0.992
2	0.958	0.958	1	1	1	0.983
3	1	1	1	1	1	1
4	1	1	1	1	1	1
5	1	1	1	1	1	1
6	1	1	1	1	1	1

CVI (Content Validity Index): the number of expert panel members who rated the item (index of 3 or 4) divided by the total number of expert panel members (*n* = 8); CVI is higher than 0.79, the item is appropriate. If between 0.70 and 0.79, the item needs revision. If the score is less than 0.70, the item is eliminated.

## Data Availability

The datasets used and/or analyzed during the current study are available from the corresponding author upon reasonable request.

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
