# Peer review of "Development and Validation of Safe Motherhood-Accessible Resilience Training (SM-ART) Intervention to Improve Perinatal Mental Health"

_ijerph, 2023, doi:10.3390/ijerph20085517_

Round 1

Reviewer 1 Report

The manuscript reports on a development of an intervention for women in the perinatal period and its assessment. The literature review is quite good and comprehensive, and the aims of the study are clear. It seems to be an important new tool for Pakistani women, but its external validity for other population is unknown. The main problem is that as an empirical study, it doesn’t describe in enough detail the content of the interviews with both pregnant women and specialist. So, we are left with description of the “solution” without the opportunity to learn about the problem.

I have some additional reflections:

  1. Resilience is a highly studied concept with a solid theoretical background. This is not reflected at all in the manuscript. A more theoretical review is thus needed in the Introduction, and a deeper connection to theory should be added to the Discussion.
  2. Line 106 – I don’t understand why all the interviews had to be translated to English from Urdu and then back-translated to Urdu. How does that process validate the responses? Please clarify.
  3. The description of the methodology seems systematic but lacks enough depth on contents, thus it is descriptive of the process but not on the exact content of the planning, intervention and assessment.
  4. It is not clear how novel is this intervention in the sense of comparison to existing resilience interventions. The authors should extend discussion on this issue.

Reviewer 2 Report

This study is interesting, and have potential clinical aplication. Authors of this work did great job, but some weak points needs clarification. 

  1. The rationale of the study is presented vaguelly. It should be improved and revised.
  2. More details about subjects characteristics are needed.
  3. The inclusion/exclusion criteria should be described in details
  4. The methods are described vaguely and hard to follow. This paragraph should be revised deeply. 
  5. The discussion should be much more emphesized on the own results.
  6. The study limitations are lacking
  7. The clinical application of the results should be clearly presented. 
  8.  

Reviewer 3 Report

Thank you very much for having shared with me your interesting work about, Development & Validation of Safe Motherhood-Accessible Resilience Training (SM-ART) Intervention to Improve Perinatal Mental Health.

I have carefully revised the manuscript and suggest to authors substantial changes as follow:

43-45 you need describe in detail (via literature) what mental health consequences have been associated in children development since pregnant women were no-resilience. It is hardly tied with your research scope.

48-50 authors reported APGAR scores such as outcome measure. What other scales/instruments/questionnaires/scores have been negatively associated with depression in pregrants?

58-59 I suggest adding certified websites on mental health of pregnant women and health of children.

60-74 authors report coping strategies during motherhood decrease stressors. Nevertheless, authors should include literature confirming this hypothesis and related outcomes. Similarly, subsequently when they indicate CBT or positive interventions.

93  the phrase “comprehensive review” is not appropriate here.

94 As the enrolled experts, authors need include socio and persona information table of the enrolled women (first sample).

Phase I. Authors should include the content of the interview provided to pregnant, as well as the methodology of its structure (psychometric data). It is not clear if this interview was in line with previous literature.

116 The authors should show in the table the results of the phase I

119 Please you describe the acronyms

120 I do not understand if the author describes the interview or other, please clarify.

phase II In this section you need including the table 1. These modules are contained in similar researches?

155 “Review” I do not read any citations.

174-180 Please move this section in method section, it would be more appropriated.

191 Who have designed the pictures? This can be interesting.

191-206 Please move this section in method section, it would be more appropriated.

199 “ delivery”. Throughout the manuscript it is not clear if the training has been provided or not. Please clarify the method, the training and the result.

Discussion: you need including the method limitations and future applications (future studies) of your training. The modules of the training have an impact individually or when provided ensemble? Have you imagined other stakeholders for this training? What other measure/questionnaire you could include in the method? In the literature, you have great choice in the mental health measures such as Coping, Depression, Resilience, Self-esteem, you should include these outcomes surely.

Concluding, I need revising newly this manuscript after major changes (citations, paragraphs, tables, result, discussion).

Good Luck, very interesting topic

XY

Round 2

Reviewer 1 Report

N/A

Reviewer 2 Report

Authors addressed all my comments sufficiently and strongly improved the manuscript. I recommend accept it for publication

Reviewer 3 Report

Thank you once again, having changed more sections of the manuscript.

suggest to you other changes still.

43-45 “Perinatal mental illnesses are strongly associated with increased rates of infant, and stunting[9-13]”. Please, rewrite this sentence in more clear way.

152-153 In my opinion you need describe as you have extracted data from the interviews (you have added the content, but not the scores/methodology) have you utilized content analysis measures? the TABLE 1 should contain the characteristics of the sample 1 (women) and the percentage/frequency/rate of response replacing the data of the experts (please, put them only in the text).

170 “Therefore, the research team members (Shireen Shahzad Bhamani, An Sofie Van Parys, and David Arthur)” I don't recommend reporting your name while you could add the expertise.

131 “All women were married and 18 years or older, with 12 weeks and above gestational” please, add if the women have received diagnosed (Anxiety, depression, bipolar...) and if they have received psychotherapy training, and their previous pregnancy (Pregnancy interruptions...) , other than if they had other children.

260 … “thatand”…?

316, Table 5. Demographic Data of Content Experts, I suggest to you adding the information of eight experts in the section and eliminate the table cause is superfluous, as well as, the content of their feedbacks. (329-375). The content of the expert feedback should be included in method since you successively have elaborated data. Finally, I suggest to you cancelled. Also, TABLE 1 you need including the report of the women while the socio-demographic data of the expert could be maintained in the related section (only text).

In sum up, there are different formal mistakes when you using.. [ please check them throughout the manuscript.

Thank you in advance for your new revisions.

Cordially,

XY
